# Nitrogen isotopes reveal a particulate-matter driven biogeochemical reactor in a temperate estuary

Kirstin Dähnke[1], Tina Sanders[1], Yoana Voynova[1], and Scott D. Wankel[2]

[1]Institute of Carbon Cycles, Helmholtz-Zentrum Hereon, Geesthacht, 21502, Germany
[2]Woods Hole Oceanographic Institution, Woods Hole, 360 Woods Hole Rd, MA 02543, USA

*Correspondence to*: Kirstin Dähnke (Kirstin.daehnke@hereon.de)

**Abstract.** Estuaries and rivers are important biogeochemical reactors that act to modify the loads and composition of nutrients in the coastal zone. In a case study during July 2013, we sampled an 80km transect along the Elbe estuary under low-oxygen conditions. To better elucidate specific mechanisms of estuarine nitrogen processing, we tracked the evolution of the stable isotopic composition of nitrate, nitrite, particulate matter, and ammonium through the water column. This approach allowed constraint of the in-situ isotope effects of ammonium and nitrite oxidation and of remineralization at the reach scale. The isotope effects of nitrite oxidation and ammonium oxidation were consistent with pure culture assessments. We found that the estuarine nitrogen budget of the Elbe estuary is governed by settling, resuspension, and remineralization of particulate matter, and further used our stable isotope data to evaluate sources and sinks of nitrogen in the Elbe estuary through an isotope mass balance approach. We find that the reactivity of particulate matter, through its remineralization in the estuary, is the main control on nitrogen isotope dynamics of inorganic nitrogen species. Moreover, while underscoring this role of particulate matter delivery and reactivity, the isotope mass balance also indicated additional sinks of reactive nitrogen, such as possible denitrification of water column nitrate in the intensively dredged and deep Hamburg harbor region.

## Introduction

Estuaries are biogeochemical hotspots of elemental cycling and can reduce contaminant and nutrient loads, thus playing a major role in the transfer of terrestrial nutrients and organic matter to the coastal zone. With growing use of synthetic fertilizers over the last 150 years, river-borne loads of nitrogen (N) to the coastal zone have greatly increased (Galloway et al., 2003). Much research over the past several decades has focused on understanding the role of estuaries in mitigating these surplus nutrient inputs (Bonaglia et al., 2014; Eyre and Balls, 1999; Garnier et al., 2010; Pastuszak et al., 2005; van Beusekom and de Jonge, 1998). As an example, the estuary of the Elbe River, one of the largest rivers discharging into the North Sea has served to remove substantial inputs of surplus reactive nitrogen (Schröder et al., 1996). With improved wastewater management and control of fertilizer application within the watershed, however, the overall riverine load of reactive nitrogen has decreased. Although direct ammonium loading was reduced by the early 1990s (Bergemann and Gaumert, 2010), nitrate loads have only gradually declined (Pätsch et al., 2010). Nitrate sources are primarily linked to agricultural land-use in the catchment (Johannsen et al., 2008; Radach and Pätsch, 2007). Along with an overall improvement of estuarine oxygen levels in comparison to the early 1990s after the fall of the Iron Curtain (Spiekermann, 2021), nitrification, the microbial oxidation of ammonium to nitrite and nitrate, gained in importance in the N budget (Dähnke et al., 2008). Today, oxygen depletion occurs only regionally in summer in the harbor region (Schöl et al., 2014), where

nitrification is most active and at times even doubles the nitrate load of the agriculturally impacted river (Sanders et al., 2018). As a consequence, nitrate pollution remains problematic – and improving our understanding of the factors regulating its microbial production and/or removal are critical.

The nitrogen load and composition in rivers and estuaries is heavily modified by microbial transformation processes, including those involving net inputs or outputs (N fixation, denitrification) as well as those that involve more internal cycling transformations (e.g., assimilation, nitrification, remineralization). However, constraining this microbial turnover and recycling of nitrogen can be especially difficult using concentration measurements alone. As a complementary and powerful tool, changes in the isotope composition of reactive N species have been widely leveraged to provide new insights into biological N turnover in estuarine (Dähnke et al., 2008; Middelburg and Nieuwenhuize, 2001; Sebilo et al., 2006), coastal (Dähnke et al., 2010; Wankel et al., 2007) and open ocean systems (Stephens et al., 2020). This utility has been especially powerful for combined analyses of $\delta^{15}N_{NO3}$ and $\delta^{18}O_{NO3}$ in evaluation of biological N turnover across a variety of natural systems (Buchwald et al., 2015; Dähnke et al., 2010; Sigman et al., 2009; Wankel et al., 2006). Biological N transformations give rise to isotopic fractionation, leading to measurable shifts in the isotopic ratios of substrate and product pools (Kendall, 1998; Mariotti et al., 1982; Sebilo et al., 2006). These isotopic shifts can be compared with fractionation factors determined from pure culture studies (e.g. Buchwald et al., 2012; Casciotti et al., 2003; Granger et al., 2004; Jacob et al., 2018). For example, the $^{15}N$ fractionation factor (or isotope effect, $^{15}\varepsilon$) for ammonia oxidation in bacterial and archaeal pure cultures ranges from -13 to -41‰ (Casciotti et al., 2003; Mariotti et al., 1981; Santoro and Casciotti, 2011), whereas nitrite oxidation has a fractionation factor from +10 to +12‰, reflecting a unique inverse isotope effect (Casciotti, 2009; Jacob et al., 2018). The fractionation factor imparted during remineralization / ammonification of organic nitrogen (into ammonia) is generally small, with estimates ranging from ~0‰ (Brandes and Devol, 2002; Kendall, 1998) to -2.5‰ (Möbius, 2013). Nitrate can be removed by denitrification in anoxic regions. During sedimentary denitrification, no isotope effect is expressed as rates are diffusion limited (Brandes and Devol, 1997). In contrast, denitrification occurring in the water column exhibits an isotope effect of up to 30‰ (e.g., Brandes et al., 1998). In natural marine environments, the use of stable of stable isotope nitrogen dynamics to elucidate N-cycling (e.g., Gaye et al., 2013; Granger et al., 2013; Sigman et al., 2009), is often restricted to individual N species, such as nitrate or particulate nitrogen, simply because low concentrations of other N–bearing compounds preclude their isotopic measurement.

In many environments, the influence of nitrification and denitrification may be closely coupled or co-occur (e.g. Brase et al., 2017; Granger and Wankel, 2016). Here in the Elbe estuary, we leverage observed patterns in stable isotopes and nitrogen budgets under an intense summer oxygen depletion period to provide perspective on the dynamics of both processes. The spatial extent of nitrite and ammonium accumulation during this period offers a unique constraint on in-situ isotope effects of nitrification in a natural system. We further relate this detailed assessment of nutrient and isotope composition to factors that control biological nutrient turnover in summer in the estuary.

## 2. Material and Methods

### 2.1 Study site

The Elbe Estuary extends from the weir in Geesthacht at stream km 586 (measured from the German-Czech border) to Cuxhaven at stream km 714, where the estuary opens to the Southern North Sea. The Elbe River catchment is

dominated by agricultural land-use and is inhabited by ~25 million people (Lozán and Kausch, 1996). The most important nitrogen inputs originate from agricultural sources in the upper and middle reaches of the Elbe (Hofmann et al., 2005; Johannsen et al., 2008) and reach the Elbe via diffuse inputs to its tributary rivers. Point-sources play a subordinate role (Hofmann et al., 2005). The Elbe River is the most important source of reactive nitrogen to the Southern North Sea, with an average annual load of ~ 75 kt between 2012 and 2016 (Umweltbundesamt, 2021). Riverine nutrients have been reduced by ~50% since the 1990s, but the Southern North Sea is still regarded as a problem area of eutrophication (OSPAR Commission 2008). The tidal part of the Elbe estuary also serves as an important shipping fairway. Continuous maintenance dredging enables access for large container ships to the harbor of Hamburg, the third largest harbor in Europe.

## 2.2 Sampling

In July 2013, we sampled the water column of the estuary from the mouth to stream km 610, upstream of the city of Hamburg (Figure 1) with the R/V *Ludwig Prandtl*. Surface water samples were collected with an on-board diaphragm pump via the vessel's water inlet at a depth of 2 m, and turbidity, salinity, pH and dissolved oxygen were measured continuously using an in-situ FerryBox system (Petersen et al., 2011). Chlorophyll is reported based on fluorescence measurements from a SCUFA Fluorometer (Turner Designs, San Jose, CA, USA), integrated in the Ferrybox. No final calibration was performed, and chlorophyll fluorescence data thus can only be used to assess relative changes along the transect. This SCUFA instrument also reported turbidity, and was calibrated against FNU standards in the laboratory prior to the cruise. Dissolved oxygen in the FerryBox was measured with an optode (Aanderaa Data Instruments AS, Bergen, Norway).

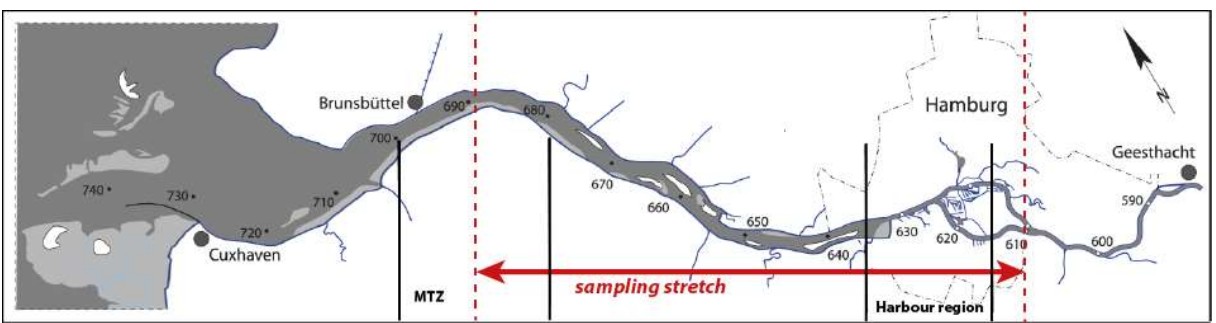

**Figure 1: The Elbe estuary. The stretch of the sampling transect is indicated in red. Vertical lines black lines show the location of the Hamburg harbor area (stream km 615 – 635) and of the estuarine maximum turbidity zone (stream km 680 – 700).**

Water samples were immediately filtered through pre-combusted (4 hrs, 450°C) GF/F filters (0.7 µm) and stored in acid-washed (10% HCl overnight) PE bottles before they were frozen at −18°C for later analyses of nutrient concentration and dissolved inorganic nitrogen (DIN) stable isotope composition. Filter samples for suspended matter were dried at 50°C directly after sampling and were then stored frozen. Nitrification rates were immediately determined on board using fresh and unfiltered water samples (see section 2.5 below).

### 2.3 Nutrient analysis

Duplicates of all filtered water samples were analyzed for concentrations of ammonium, nitrite und nitrate, using an automated continuous flow system (AA3, Seal Analytical, Germany) and standard colorimetric techniques (Hansen and Koroleff, 2007). Detection limits were 1 µmol $L^{-1}$ for nitrate, 0.5 µmol $L^{-1}$ for nitrite and 0.5 µmol $L^{-1}$ for ammonium. In subsamples of nitrification rate measurements, ammonium was analyzed manually based on the same colorimetric techniques. Due to restrictions in sample volume, nitrite and nitrate concentrations in these incubation samples were analyzed in replicate by HPLC (Jasco 980, Jasco, Germany) (Meincke et al., 1992).

### 2.4 Analyses of suspended particulate matter

Suspended particulate matter (SPM) was analyzed for its carbon and nitrogen content, and isotopes thereof. Sensor data of chlorophyll fluorescence were further used as an indicator of suspended matter quality. Total particulate carbon (POC) and total nitrogen (TN) concentrations (weight%) of SPM were quantified using a Thermo Flash EA 1112 (Thermo, Germany). The precision of the measurement was 0.05% for carbon and 0.005% for nitrogen, measurements were calibrated with a certified acetanilide standard (IVA Analysentechnik, Germany). $\delta^{15}N$ values of SPM were measured with an isotope ratio mass spectrometer (Delta Plus XP, Thermo, Germany) after combustion in an elemental analyzer (Flash EA 1112, Thermo, Germny). IAEA-N1, IAEA-N2 and an additional certified sediment standard (IVA Analysentechnik, Germany) were measured every eight to ten samples during each run to assure measurement quality and assess long-term analytical performance. A single-point calibration was performed with IAEA-N1 with an assigned $\delta^{15}N$ value of 0.4‰ (Bahlmann et al., 2010). The standard deviation for IAEA-N1 isotope analyses was less than 0.2‰.

### 2.5 Nitrification rates

Nitrification rates were measured following a modified version of the ISO/DIN 15685:2001 standard test (DIN15685 2001). The method is described in detail in Sanders and Laanbroek (2018). In short, 50 ml of river water were incubated in 100 mL glass bottles with open caps in the dark at room temperature (22±2°C, i.e., close to the average water temperature of ~24°C) for up to 14 days (Sanders and Laanbroek, 2018). Up to 5 sub-samples per week were taken, centrifuged (15 min, 13.000 g), and nitrite/nitrate concentrations were determined immediately by HPLC. Incubations were continued until the nitrite and nitrate concentrations in the samples were stable, indicating that remineralization of labile organic matter was complete. This approach is used to address potential nitrification that is possibly fueled by organic matter in the given water sample. Details are discussed in Sanders and Laanbroek (2018).

In total, 4 replicate analyses were conducted; in 2 bottles 0.5 mM potassium chlorate was added to inhibit nitrite oxidation (Belser and Mays, 1980), and 2 unamended replicates were measured in addition. Process rates were calculated by plotting concentration change over time, with the steepest portion of the slope corresponding to the rate. We found that the slope of amended and unamended incubations did not differ significantly between the measurements, indicateing that the rate limiting step was ammonia oxidation. Thus, the slopes of nitrite accumulation (amended treatments) and nitrate accumulation (unamended treatments) are comparable, and nitrification rates are presented as the mean values of all 4 incubations.

### 2.6 Stable Isotope analysis

#### 2.6.1 Nitrate isotopes

All water samples were analyzed for isotopic composition of nitrate ($\delta^{15}N_{NO3}$ and $\delta^{18}O_{NO3}$) using the denitrifier method (Casciotti et al., 2002; Sigman et al., 2001), wherein nitrate and nitrite are quantitatively converted to nitrous oxide ($N_2O$) by denitrifying bacteria (*Pseudomonas aureofaciens*, ATCC#13985) lacking $N_2O$ reductase. Injected sample volumes were adjusted to achieve a sample size of 20 nmol of generated $N_2O$. $N_2O$ was extracted from the sample vials by a flow of ultra-high purity helium and measured with a GasBench II (Thermo, Germany), coupled to an isotope ratio mass spectrometer (Delta Plus XP, Thermo, Germany). With each batch of samples, two international standards (USGS34: $\delta^{15}N$: -1.8‰, $\delta^{18}O$: -27.9‰; IAEA-NO$_3^-$: $\delta^{15}N$: +4.7‰, $\delta^{18}O$: +25.6‰) and an internal standard were also measured for standardization. Standard deviation of samples and standards was <0.2‰ for $\delta^{15}N_{NO3}$ (n = 4) and <0.5‰ for $\delta^{18}O_{NO3}$ (n=4). In samples with elevated nitrite concentration, $\delta^{15}N_{NO2}$ was analyzed separately, and we present $\delta^{15}N_{NO3}$ values that were mass-balance corrected for the contribution of nitrite.

#### 2.6.2 Nitrite isotopes

During the sampling campaign, high nitrite and ammonium levels in some samples allowed for isotope analysis. $\delta^{15}N_{NO2}$ was analyzed in samples with $[NO2^-] \geq 2$ µmol L$^{-1}$ according to previously described protocols (Böhlke et al., 2007). A culture of *Stenotrophomonas nitritireducens* is used to selectively reduce nitrite to $N_2O$. For calibration, we used two in-house standards with known $\delta^{15}N_{NO2}$ values of -83.3 ±0.2‰ and +27.6 ±0.2‰ vs. Air-$N_2$, determined independently via EA-IRMS analysis. For $\delta^{15}N_{NO2}$, the standard deviation of our measurements was <0.3‰ (n=4).

#### 2.6.3 Ammonium isotopes

$\delta^{15}N$ in ammonium ($\delta^{15}N_{NH4}$) was analyzed based on the hypobromite method (Zhang et al., 2007). In brief, any sample nitrite was removed by addition of sulfamic acid (Granger and Sigman, 2009), followed by conversion of ammonium to nitrite by hypobromite oxidation. Nitrite was then chemically converted to $N_2O$ via addition of acetic acid buffered sodium azide solution and was measured by purge and trap as with nitrite and nitrate isotope samples (McIlvin and Altabet, 2005). The minimum ammonium concentration required for isotope analyses in our setup is 2 µmol L$^{-1}$. With each batch of samples, three international ammonium standards (USGS25: $\delta^{15}N$: -30.4‰, USGS26: $\delta^{15}N$ +53.8‰, IAEA N1: $\delta^{15}N$ +0.4) were run for calibration, and an internal standard was used for quality control. The standard deviation of samples and standards was <0.5‰ for $\delta^{15}N_{NH4}$.

### 2.7 Estimation of isotope effects for nitrite oxidation and ammonium oxidation

To determine the isotope effects of N cycling processes in the Elbe Estuary, we conceptualized the data using an open-system model, where the substrate is continuously supplied and partially consumed. This results in a linear relationship between f (the fraction of the remaining substrate, $f = ([C]/[C_{initial}])$) and the isotope values of the substrate, where the slope of the regression line represents the isotope effect (Sigman et al., 2009), calculated following Eq. 1:

$$\varepsilon_{substrate} = \frac{\delta^{15}N_{substrate} - \delta^{15}N_{initial}}{(1-f)} \tag{1}$$

$\delta^{15}N_{substrate}$, $\delta^{15}N_{product}$, and $\delta^{15}N_{initial}$ denote $\delta^{15}N$ values of the substrate and product at the time of sampling and the initial value, and f is the remaining fraction of substrate. For ammonium and nitrite consuming reactions, isotope effects were calculated only along flow sections in which substrate concentration decreased. For ammonium, this corresponded to stream km 626 to 639. For nitrite, we used data from between stream km 641 and 656. In this section, ammonium concentration is already low (maximum of 3.5 µmol L$^{-1}$), and we assume that the relative impact of ammonia oxidation on the nitrite pool is negligible.

The isotope effect of remineralization of particulate nitrogen ($^{15}\epsilon_{remin}$) was estimated based on changes in the $\delta^{15}N$ and in the N-content of particulate matter (Möbius, 2013) over the entire transect.

## 3. Results

### 3.1 Biogeochemistry and nutrients along the estuary

To unravel estuarine biogeochemical cycling, we regard hydrological parameters and nutrient trends in the freshwater section of the estuary, starting in the shallow freshwater section, through the dredged harbor area, up to the onset of the salinity gradient, when mixing with low concentration seawater begins. Water temperature in the estuary was exceptionally elevated for this time of year, with temperatures up to 24°C at the uppermost sampling site. Oxygen saturation dropped from ~100% saturation at stream km 610 to minimum values of 32 – 34% in the Hamburg harbor region. Downstream of the harbor region, oxygen saturation increased again to final values near 100% in the lower estuary (Fig. 2). Turbidity and suspended particulate matter concentration were quite variable along the sampling section (Table S1). However, both showed relatively high values at the uppermost stations that dropped at the entrance to the harbor region, followed by a peak around stream km 639. A small peak also occurred near the estuarine turbidity maximum at stream km 670. Chlorophyll fluorescence rapidly also dropped at the entrance to the harbor region. A transient maximum appeared around stream km 648, slightly after maximum values for turbidity. No notable change in chlorophyll fluorescence was noted at the estuarine turbidity maximum. The molar C / N ratio of particulate matter exhibited values of ~6.6 in the incoming water, with an increase in the harbor basin to ~7.4, and a decrease towards 6.7 in the transient chlorophyll maximum at stream km 648. In the lower regions of the estuary, C / N increased to values near 10 (Figure 2 and Table S1).

Generally, nutrient concentrations in the freshwater estuary were dynamic, suggesting active uptake and/or recycling (Fig. 2). Concentrations of the dominant DIN species nitrate ranged from 87 µmol L$^{-1}$ at stream km 610, upstream of the harbor region, and increased to a maximum of 152 µmol L$^{-1}$ just before the onset of the salinity gradient at stream km 690. Ammonium and nitrite were present in lower concentrations. The ammonium concentration at the upstream station entering the harbor region was 2 µmol L$^{-1}$. Ammonium then peaked in the harbor region (stream km 615 – 641) with a concentration of 21 µmol L$^{-1}$ at stream km 626. Just downstream, the ammonium peak was followed by a peak of nitrite between stream km 620 and 656, with a maximum concentration of 13 µmol L$^{-1}$ at stream km 641. Silicate concentrations also increased from the upstream station (10 µmol L$^{-1}$) to a peak at stream km 641 (53 µmol L$^{-1}$). Further downstream, silicate concentrations then decreased to less than 5 µmol L$^{-1}$, with a small peak around stream km 670, at the beginning of the estuarine maximum turbidity zone (Figure 2). Phosphate concentration was at its minimum upstream of the harbor region, where the water is shallow. From 0.2 µmol L$^{-1}$ at stream km 618, concentrations increased to < 2 µmol L$^{-1}$ (1.7 +/- 0.6) in the Hamburg harbor region and then remained relatively stable throughout the estuary.

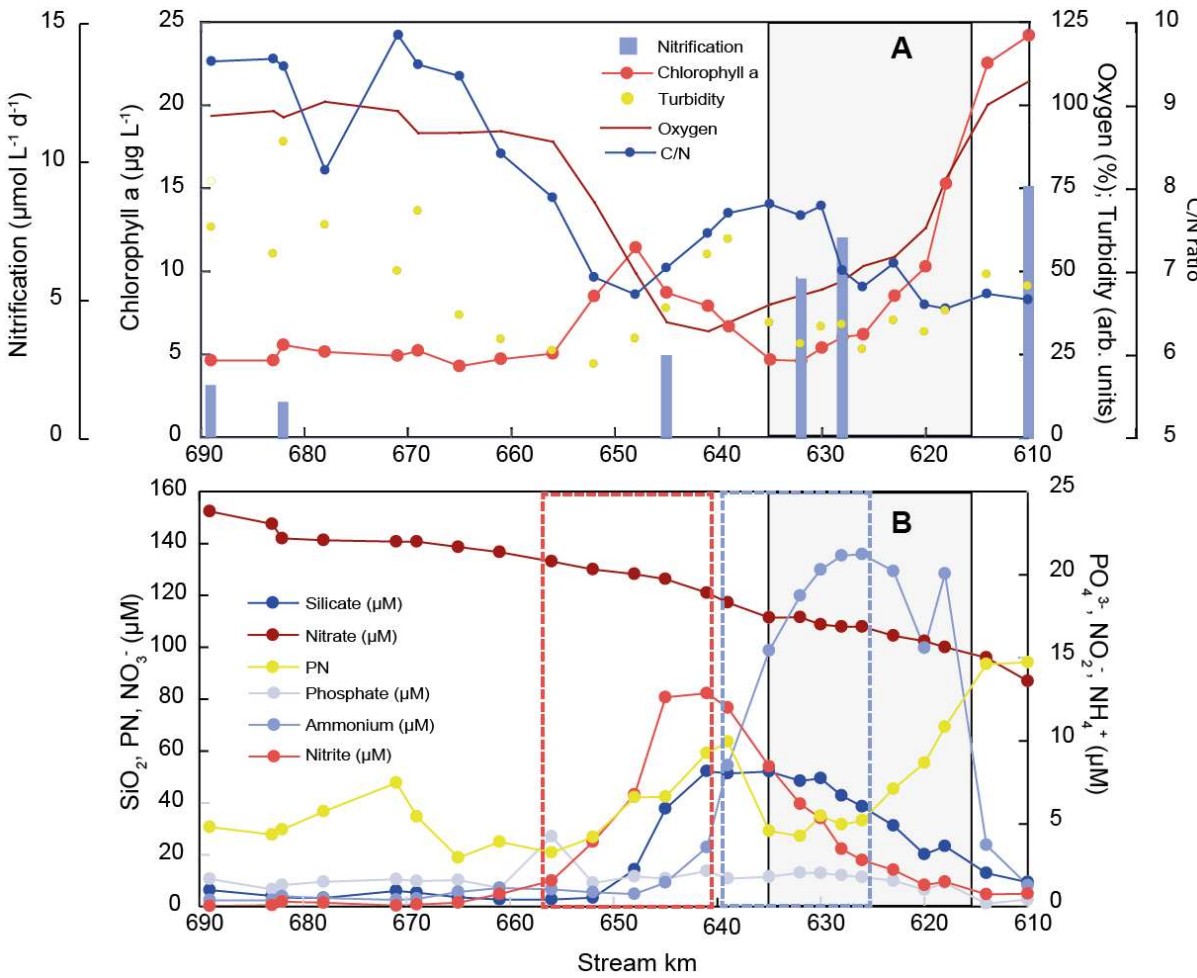

**Figure 2: Nutrient concentration, particulate matter composition, biogeochemical trends and nitrification rates along the Elbe Estuary. The harbor region is indicated by shading, and the dashed blue and red boxes show the stretches that were used for the estimation of the isotope effects of ammonium and nitrite removal, respectively.**

### 3.2 Nitrification

Nitrification rates in July 2013 were highest in the upper part of the estuary, including the Hamburg harbor region (Elbe-km 618 to 635) with a nitrate production of up to 10 µmol $L^{-1}$ $d^{-1}$ (Fig. 2A). Nitrification rate decreased downstream of the harbor region, reaching minimum values of ~2 µmol $L^{-1}$ $d^{-1}$. Rates were strongly positively correlated to %N and %C in suspended matter ($r^2 = 0.91$ and $0.95$, respectively), and negatively correlated to $\delta^{15}N$ of SPM. Lower $\delta^{15}N_{SPM}$ values coincided with high nitrification rates (not shown). Nitrification rates were not correlated with dissolved oxygen concentration or SPM concentration.

### 3.3 Isotopic changes along the estuary

Stable isotope ratios of nitrate and suspended particulate nitrogen varied uniformly along the entire estuary (Fig. 3). Nitrate $\delta^{15}N$ and $\delta^{18}O$ values were elevated entering the estuary (+19‰ and +10‰ for $\delta^{15}N$ and $\delta^{18}O$, respectively), decreasing along the estuary to final values of +11 and +3‰, respectively, around stream km 656 (Fig. 3). Downstream of this point, nitrate stable isotopes exhibited conservative mixing patterns (data not shown).

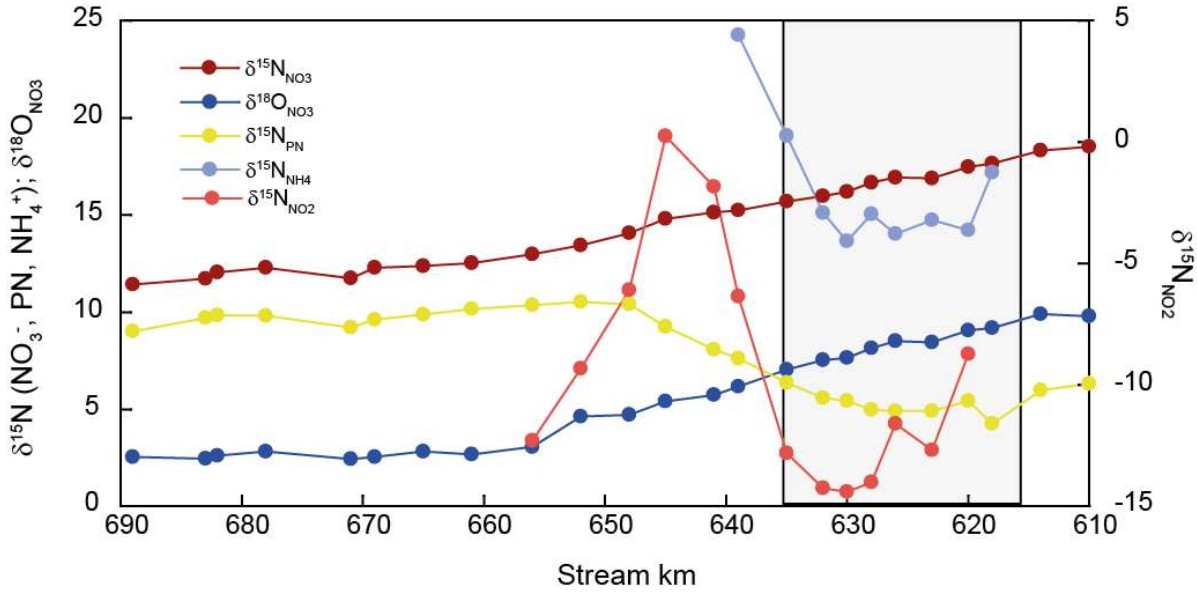

**Figure 3: Isotopic changes of DIN and PN along the estuary. The harbor region is indicated by shading.**

Nitrogen isotope composition of the suspended particulate matter ($\delta^{15}$N-SPM) was most dynamic in the harbor region. $\delta^{15}$N values of SPM at the entrance of the sampling transect were +6.3‰, and the isotope value dropped to a minimum of +4.9‰ at the beginning of the harbor region. Between stream km 628 and 648, $\delta^{15}$N-SPM swiftly increased to +10.4‰, and then remained relatively stable throughout the rest of the estuary, with a slight decrease to +9.0‰ towards the onset of the salinity gradient.

Between stream km 620 and 656, where concentrations were high enough to analyze nitrite isotopic composition, $\delta^{15}$N values exhibited large variations. At the entrance to the harbor, $\delta^{15}$N-NO$_2$ was -9‰, before dropping to a minimum of $\sim$ -15‰, and then increasing, in parallel with increasing concentration, to a maximum of $\sim$0‰ at stream km 641. The subsequent decrease in [NO$_2^-$] was accompanied by decreasing $\delta^{15}$N-NO$_2$ values that reached a minimum of -12‰ at stream km 656. Using data from this reach, an inverse isotope effect of +12.9 ± 1.3 ‰

was estimated for nitrite oxidation (Fig. 4). Ammonium $\delta^{15}$N values, measured between km 618 and 639 (Fig. 3) were negatively correlated with ammonium concentration. At the entrance of the harbor, $\delta^{15}$N-NH4 was +17.7‰, decreasing to $\sim$ +15‰ in the harbor region before increasing to +24‰ as ammonium concentration decreased (stream km 635 to 639). Using data from this reach, the isotope effect associated with this ammonium removal was estimated as $\varepsilon_{amm}$ = -17.4 ± 1.7‰. To calculate the isotope effect of remineralization, we used an open system

approach that was based on $\delta^{15}$N of suspended particulate matter (ln(N%), Möbius 2013) across the entire transect. This approach yielded an isotope effect of $\varepsilon_{remin}$= - 4.4 ± 0.8‰ (Fig. 4).

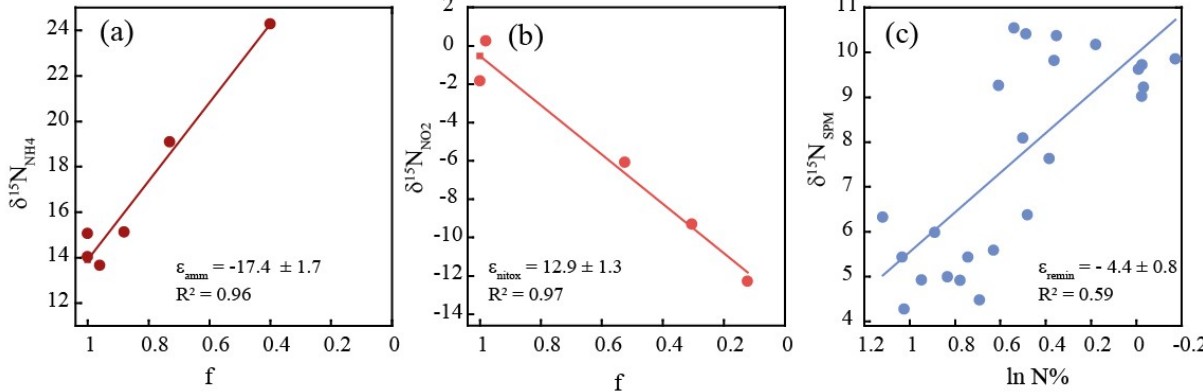

Figure 4: Overview of calculated fractionation factors for turnover of (a), ammonium, (b), nitrite, and (c), particulate nitrogen along the respective Elbe stretch. The initial f denotes the remaining fraction of the respective substrate. See sections 2.7and 3.3 for details of the calculation.

## 4. Discussion

### 4.1 Deriving N-cycling from isotope effects

In this study, we captured a rare event in the Elbe estuary in which the isotopic characterization of transient N-cycle intermediates that accumulated after a summer flood in the same year (Jacob et al., 2016;Voynova et al., 2017) could be leveraged to provide insight into riverine process dynamics. The isotope and concentration changes in our dataset underscore the high degree of biogeochemical activity in the harbor region – as has been noted previously (Brase et al., 2017; Sanders et al., 2018). Nitrification rates measured in this study were similarly distributed to, if slightly higher than, those previously reported (Sanders et al., 2018). Notably, however, the distribution of nitrification rates along the estuary matches the succession of transient ammonium and nitrite accumulation through the harbor basin. The spatial extent of ammonium and nitrite accumulation during our cruise enabled us to compute individual isotope effects for N-turnover processes. With our assessment of isotope effects here, we aim to more fully characterize these dominant turnover processes.

Notably, nitrate isotope composition only faintly reflects nitrate regeneration dynamics in the estuary (Fig. 2A, Fig. 3). Recent work evaluating nitrate production in the estuary noted that changes in $\delta^{18}O$ of nitrate were indicative of intense nitrification, and that newly added nitrate had an oxygen isotope signature close to that of ambient water (Sanders et al., 2018). This holds true in our dataset as well; we find a characteristic drop of oxygen isotope values in the harbor region reflecting incorporation of oxygen atoms from $H_2O$ into nitrate (Fig. 3). Nonetheless, in comparison to large isotope dynamics observed in both the nitrite and ammonium pools, changes of nitrogen isotopes in nitrate are dampened due to the large pre-existing nitrate pool. The $\delta^{18}O$ of nitrate appears more sensitive to inputs by nitrification than $\delta^{15}N$, as a result of the relative differences between the composition of new nitrate and that of the pre-existing pool.

The inverse isotope effect for nitrite oxidation (+12.9‰, stream km 641 - 656) in our study corresponds well the isotope effect that has been identified in lab experiments (Casciotti, 2009) as well as under natural conditions in the Elbe catchment (Jacob et al., 2016). We thus assume that nitrite oxidation is the main sink for nitrite in this river section, and that nitrite concentration along this stretch is largely unaffected by other sinks such as denitrification or nitrite assimilation. Between stream km 626 and 639, the estimated isotope effect for ammonium

removal of -17.4 ‰ lies towards the lower end of pure culture-based assessments of the isotope effect of ammonia oxidizing bacteria (-14 to -38, Casciotti et al., 2003). If ammonia oxidation represents the sole removal mechanism, the steady-state difference between the isotope signature of ammonium and nitrite ($\delta^{15}N_{NH4}$ - $\delta^{15}N_{NO2}$) where $NH_4^+$ is decreasing and $NO_2^-$ is accumulating, should approximately equal the isotope effect calculated based on Eq 1. Interestingly, this is not the case, $\delta^{15}N_{NH4}$ - $\delta^{15}N_{NO2}$ is 28.8 ± 2.3‰, which is larger than the computed isotope effect of ammonium oxidation. This suggests that another ammonium removal process with a lower isotope effect occurs in parallel. A likely candidate is ammonium assimilation by phytoplankton, which would fit with increasing chlorophyll fluorescence along this stretch (Figure 2).

The isotope effect of ammonium assimilation is highly variable, (e.g., Hoch et al., 1992; Pennock et al., 1996; Waser et al., 1998), but usually below 10‰ in estuaries (York et al., 2007) and references therein. For a rough calculation, if we assume a moderate isotope effect of 8‰ for ammonium assimilation (cf. Wada and Hattori, 1978; York et al., 2007), then approximately 55% of ammonium uptake in the water column is due to assimilation, while approximately 45% is oxidized to nitrite. Within the given uncertainty for the isotope effect of ammonium assimilation, this fits relatively well with the ratio of the ammonium and nitrite peaks (~ 2 to 1) in the estuary. While we cannot precisely determine the exact proportion of both ammonium sinks, it seems evident that ammonium assimilation is a relevant ammonium sink in addition to nitrification. This seems plausible, since phytoplankton prefers ammonium over nitrate as a nitrogen source (Dortch et al., 1991).

The calculated the isotope effect of remineralization along the transect was $\varepsilon^{15}_{remin}$ = -4.4‰. This isotope effect is higher than the usual assumption for remineralization in sediments, which ranges from insignificant to ~2‰, (Brandes and Devol, 1997; Mobius et al., 2010). We cannot easily explain this deviation and assume that it relates either to the highly labile and reactive nature of organic matter in our setting, which potentially may alleviate diffusion limitation during remineralization and allow the expression of a higher isotope effect. However, this explanation is speculative and should be evaluated in futrure studies. Nonetheless, the relative strength of coupling between remineralization and nitrification in the Elbe has been previously noted (Sanders et al 2018). Below we thus evaluate the role of SPM quality in the estuary and its connection to supporting N cycling.

**4.2 The role of suspended matter**

The increase in nitrate concentration, together with the transient peaks of nitrite and ammonium and their corresponding isotope dynamics, underscore the prominence of nitrification as a central N-cycling process in the harbor region. However, the strength of nitrification can only be supported by an active supply of $NH_4^+$ through the decomposition of organic matter, the reactivity of which must ultimately underlie the observed dynamics. The composition and quality of suspended matter can be very heterogeneous and is affected by remineralization, resuspension, and of course by phytoplankton growth (e.g., Middelburg, 2019;Middelburg and Herman, 2007). Our data indicate that rates of nitrification do not appear to be regulated by ammonium concentration, or by the amount of SPM in the water column, but rather scale with indicators of organic matter quality, such as the N- and C-content of organic matter.

Generally, chlorophyll and particulate nitrogen concentration co-vary along the Elbe Estuary (Fig. 2). Based on 'Redfield' stoichiometry, fresh algal material has a C / N ratio of ~ 6.6 (Redfield et al., 1963; Martiny et al., 2014; Middelburg, 2019), which is very near to values found in the uppermost samples. With ongoing degradation of organic material, this ratio increases, as easily degradable, nitrogen-rich compounds (e.g., amino acids, amino

sugars, pigments) are preferentially remineralized, while carbon-rich organic matter remains (Islam et al., 2019; Middelburg, 2019; Middelburg and Herman, 2007). This same dynamic plays out in the Elbe where SPM reactivity clearly varies throughout the estuary, approaching highest C/N ratios in the outer part of the estuary (Fig. 2, Tab. S1). Dissolved $O_2$, required for aerobic heterotrophy, reaches its minimum in the harbor region. In parallel with a slight increase of C/N in this region, chlorophyll fluorescence also decreases. This suggests that algal material from upstream is being actively remineralized (Fig. 2a). The high silicate concentration in the water column (Fig. 2b) indicates remineralization of diatom frustules and corroborates that comparatively fresh and labile suspended matter is actively remineralized in the harbor region. Downstream, SPM reactivity decreases, with lower nitrification rates, increasing C/N ratio, and only faint signs of biological activity at the maximum SPM concentration found in the maximum turbidity zone of the estuary (~stream km 670).

Overall, SPM reactivity shows a strong gradient along the estuary, but scales with nitrification rates: The N content (N%) of SPM and nitrification rates are closely correlated throughout the sampling stretch ($r^2 = 0.91$, cf. Table S1). The correlation with total SPM concentration however is not as evident: some sites of high SPM are biogeochemically active, whereas others reveal very little N-turnover. This is particularly the case in the maximum turbidity zone, where little nitrification takes place, and SPM reactivity decreases, shown by increasing C/N ratios (Fig. 2). Despite the freshwater regime, SPM in this outer section of the estuary can be of marine origin (Kappenberg and Fanger, 2007), and in our case study apparently supports very little N-turnover. In contrast, the increased depth and residence time in the harbor zone effectively promote its behavior as a bioreactor in which marine and limnic SPM are mixed (Kappenberg and Fanger, 2007). This mixing supports the co-occurrence of nitrification, denitrification, and remineralization, resulting in dramatic shifts in DIN composition.

### 4.3 Constructing an estuarine N-Budget

Changes in the water column DIN speciation and their respective isotopic composition allowed novel constraint of isotope effects and N dynamics in the estuary. Next, we construct a total nitrogen isotope mass balance to more thoroughly explore potential processes and controls on the evolution of riverine nitrogen along the estuarine reach. For this approach several key assumptions were made (the validity of which is discussed below). First, we regard the transect data as representative of a 'snapshot' in time, over which steady state can be assumed. The sampling area, while generally influenced by tidal action, lies entirely within the freshwater portion of the lower Elbe. Furthermore, over this section of the river there are no significant tributaries with respect to the water budget. Thus, our first key assumption is that changes observed in N pool size and isotopic composition are not influenced by in-mixing of other external sources, and that the primary N source is the nitrogen that enters at the top of the reach. We explicitly disregard dissolved organic nitrogen (DON) in our approach, because previous assessments were equivocal regarding the role of DON in the Elbe estuary, especially in the harbor region. A previous study (Schlarbaum et al., 2010) found a removal of ~20 μmol L$^{-1}$ from the combined reduced nitrogen pool (DON + ammonium), with an explicitly high contribution of ammonium in the harbor region. This is removal fits well with ammonium removal in our study, and we therefore assume that the fraction of reactive DON in the combined pool of reduced nitrogen pool was small. Generally, invoking a large and isotopically dynamic DON pool contrasts with the observed continuum of geochemical parameters otherwise routinely documented along this stretch of the Elbe (Amann et al., 2014; Dähnke et al., 2008; Sanders et al., 2018).

Hence, we use the sum of DIN plus particulate nitrogen in SPM (PN) to address transformations in the total nitrogen pool. For mass balance calculations, please see supplementary material (S1). We find substantial fluctuations in the total nitrogen pool over the reach (as much as 16 µmol L$^{-1}$ between two sampling points, Figure 5). However, these shifts are almost entirely encompassed by fluctuations in the concentration of PN (Figure 5 and Table S1). Notably, there were no erratic changes to the $\delta^{15}$N of the PN pool, but rather a smooth and continuous transformation over the entire 90 km reach (Figure 2). This is in line with previous measurements that suggest notable changes in turbidity over this section of the Elbe, including the turbidity maximum zone toward the lower end of the reach (Burchard et al., 2018). Thus, we assume that the large fluctuations in PN are due to settling and resuspension of fresh PN and/or patchiness in PN distribution at the surface, which was not captured during our sampling.

Resuspension and/or particle settling is not considered to be an isotopically discriminating process, so we explicitly invoke resuspension or settling only to satisfy mass balance, without any impact on the total N isotope mass balance. Thus, in practice, the PN concentration has been adjusted at each sampling point to reflect the mass balance complement of the total DIN pool – normalized to the total N entering the river at Km 610. The adjusted PN data are shown in Figure 5 (light blue).

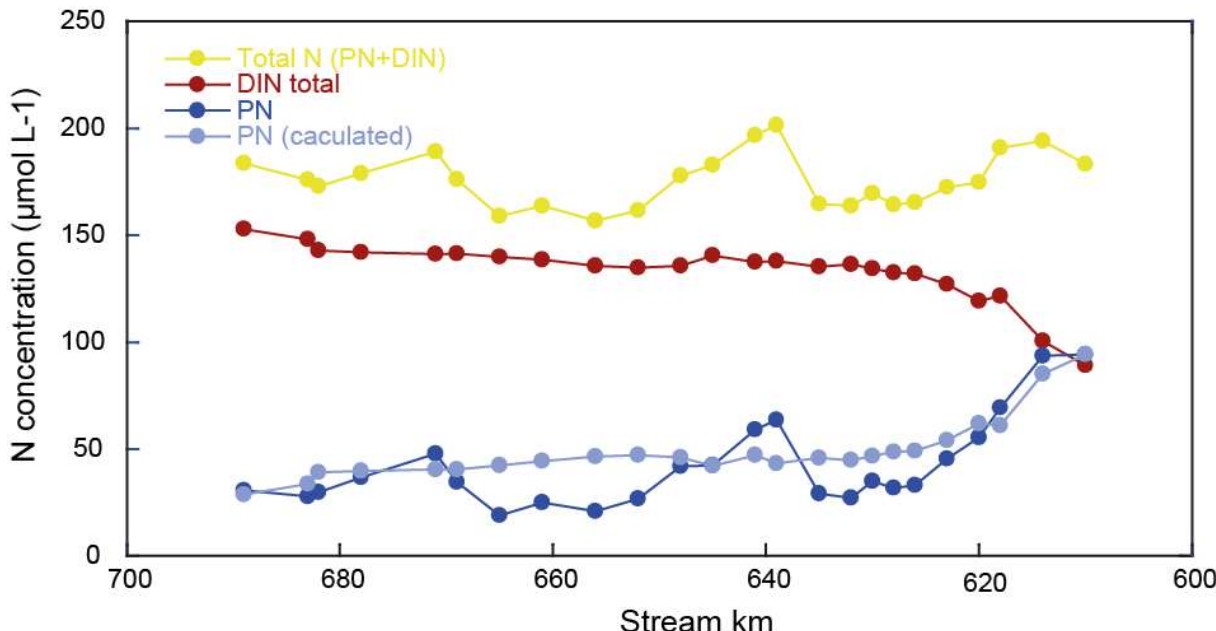

**Figure 5: N concentration along the estuary. Total N fluctuations are driven by PN fluctuations. Light blue is the calculated concentration of particulate matter that is assumed to satisfy mass balance.**

### 4.4 Implications of the N isotope mass balance

Finally, we use our mass balance calculations to investigate whether additional sources or sinks of nitrogen in the estuary may be evident that have thus far gone unrecognized. Based on the assumptions of particulate nitrogen settling and resuspension, we were able to satisfy the mass balance for each discrete sampling point. The N isotopic composition of the total N pool (Figure 5), however, exhibits distinct variations over the entire reach, which remain unexplainable due to variations of particulate matter resuspension and/or settling alone (which we suggest

do not dramatically impart shifts in isotope composition). Indeed, we contend these isotopic variations instead reflect isotopically fractionating processes or N inputs that are not addressed by the assumptions of our mass balance calculations.

Over large sections of the transect, mass balance indicates net removal of nitrogen (e.g., km 639 to km 656), and that the mass balance derived $\delta^{15}N$ values of the lost N closely resemble those of the remaining riverine N (~10‰). Thus, we might assume that sedimentary denitrification may be responsible (e.g., net N loss, with little impact on $\delta^{15}N$). However, in the upstream zone (km 618 to km 635), some of the estimated $\delta^{15}N$ values of the lost N suggest a loss of low $\delta^{15}N$ nitrogen (Figure 6). Here we suggest that water column denitrification might be a plausible loss mechanism. A loss of isotopically low $\delta^{15}N$ nitrogen could arise from denitrification of the standing $NO_2^-$ pool, which is consistently low (average of -11.8‰; Figure 3). Even a small removal flux of this very low $\delta^{15}N$ pool, could have an appreciable impact on the evolution of the $\delta^{15}N$ of the total N pool. Adopting mid-range values for the isotope effect of nitrite reduction ($^{15}\varepsilon_{NIR} \sim$ -13‰ to -16‰, (Jacob et al., 2016; Martin et al., 2019), a loss flux of less than 2% of the total N pool would be sufficient to account for the loss of low $\delta^{15}N$ nitrogen. Such a flux is plausible and in the lower range of independent estimates of present day-losses due to denitrification in the Elbe estuary (Dähnke et al., 2008; Deek et al., 2013).

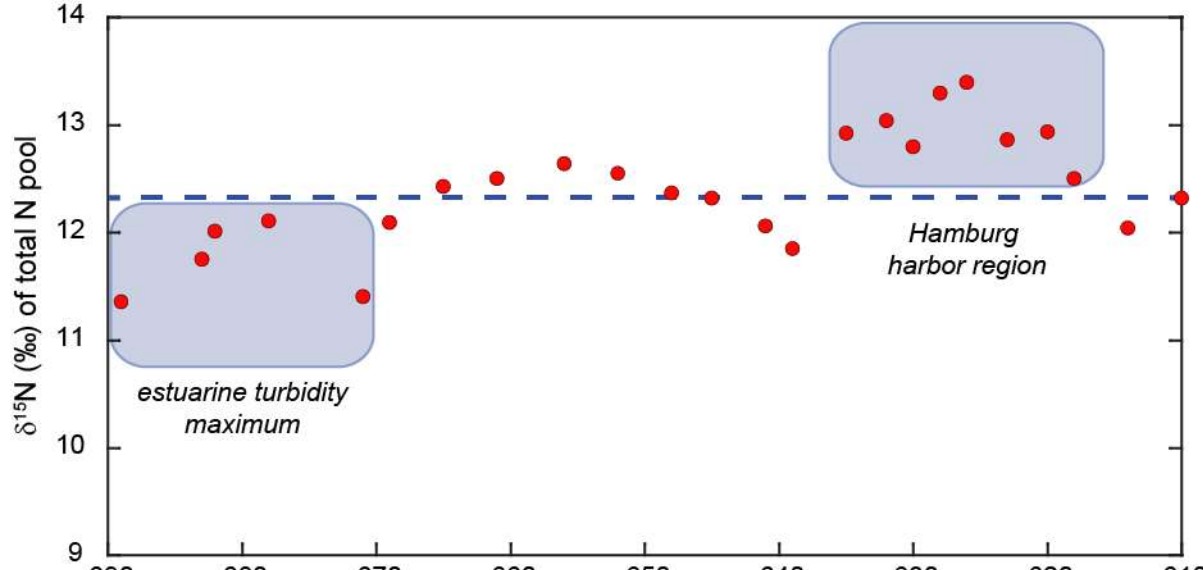

**Figure 6: Total n isotope values along our sampling stretch. The dashed blue line represents the isotope mass balance conserved input from the uppermost reach. Red dots represent the mass weighted total N pool, and deviations from the epected line indicate addition or removal of N beyond isotope-neutral processes. The harbor region appears to appears to host removal of low $\delta^{15}N$, leading to this deviation from expected isotope mass balance. As noted in the text this is consistent with water column DNF occurring in the low $O_2$ waters of the harbor. In the estuarine turbidity maximum, either high $\delta^{15}N$-nitrogen is removed, or low $\delta^{15}N$-nitrogen is added. See text, section 4.4, for details.**

Notably, this argument requires the removal of $NO_2^-$ under apparently oxic conditions ($O_2$ saturation of 40 – 63%) and relies perhaps on the existence of low oxygen conditions inside of suspended particulate or flocculate material. Anoxic nitrogen turnover inside bacterial aggregates has been demonstrated (Klawonn et al., 2015), but we do not have direct evidence of these processes in the Elbe. However, high nitrous oxide production in the harbor has been reported and could not plausibly be explained by nitrification alone, suggesting a contribution of denitrification in

the water column or sediments (Brase et al., 2017). Denitrification and nitrous oxide production in oxic waters with high suspended matter loads have been demonstrated in estuaries previously (Liu et al., 2013; Quick et al., 2019). Thus, we conclude that a contribution of denitrification in the oxic water column is at least possible in this system – and would help to satisfy the isotope mass balance in many cases.

We find a second region hosting an isotopic imbalance downstream (stream km 683 – 689). In this region, either the removal of a high $\delta^{15}N$ pool, or the addition of N with a low $\delta^{15}N$ value is required to satisfy the isotope mass balance. There are two possible explanations for this scenario. The first one is that a small addition of low $\delta^{15}N$ to the total pool is responsible for the isotopic imbalance. A potential source for this may be nitrogen fixation by azotrophic bacteria, adding low $\delta^{15}N$ to the total pool. However, we suggest that this scenario is somewhat unlikely in a eutrophic, N-loaded estuary such as the Elbe River.

Alternatively, it is possible that in this region, a small loss of nitrate occurs. While we see such a loss in the isotope mass balance further upstream leading to higher $\delta^{15}N$ values (stemming from loss of low $\delta^{15}N$ through isotope fractionation), a different mechanism may be at work here. In this section, the $\delta^{15}N$ of the $NO_3^-$ pool in this region is relatively high (~+12‰), and even a relatively minor loss of $NO_3^-$ with this elevated isotope value through sedimentary denitrification would be sufficient to lower the total N isotope composition. In fact, the loss required would not be more than 0.5 to 8.0% of the total N flux, which is in accordance with previous assessments for the Elbe (Deek et al., 2013). Thus, we conclude that denitrification may occur at low rates in the sediments in the downstream region of the estuarine turbidity maximum.

**Conclusions**

In this multi-compound isotope study, we were able to characterize the nitrogen isotope effects of nitrification and remineralization in the Elbe estuary and evaluate the role of nitrification in nitrogen turnover in this biogeochemically active region. We find that the intensity of nitrification in the estuary is coupled to remineralization and thus the reactivity of particulate matter, as indicated in its nitrogen content and C/N values. Ammonium assimilation and nitrification apparently both compete for ammonium produced from labile organic matter. Ammonium assimilation produces reactive organic matter that can in turn sustain its reactivity in the estuary. Jointly with high N loads, this organic matter also appears to fuel denitrification in the estuary through its strong stimulation of nitrification. We also find that the harbor region hosts both nitrification as well as denitrification, and speculate that even denitrification in the water column may occur in low-oxygen environments with a high load of suspended particulate matter. Overall, changes in particulate matter transport and quality appear to have profound effects on nitrogen biogeochemistry in the estuary, and  affect the relative distribution of both nitrification and denitrification. While the overall nitrogen loads of the Elbe River are decreasing, changes in river biogeochemistry clearly have dramatic local effects in the harbor region, stimulating high rates of local nutrient turnover and increased oxygen consumption.

**Data availability**

The data are available from the corresponding author upon request.

## Author contribution

Author contributions. KD, TS and SDW designed the research. TS carried out the fieldwork, and TS and YV performed the analyses. All authors jointly interpreted the data. KD, TS and SDW wrote the paper, with suggestions and additions provided by YV.

## Competing interests.

The authors declare that they have no conflict of interest.

## Acknowledgements

The authors would like to thank the captain and crew of the R/V Ludwig Prandtl for their support during the cruise. Markus Ankele is acknowledged for nutrient measurements, and we thank Justus van Beusekom for helpful comments that greatly improved this manuscript. Ji-Hyung Park and two anonymous reviewers are acknowledged for their helpful feedback that improved this manuscript greatly.

Parts of this work were supported the BMBF project Blue_Estuaries (#03F0864C) and further contribute to the cluster of excellence CLICCS (Climate, Climatic Change, and Society) of Universität Hamburg funded by the Deutsche Forschungsgemeinschaft (DFG, German Research Foundation) under Germany's Excellence Strategy – EXC 2037.

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
