# Peer review of "Nitrogen isotopes reveal a particulate-matter driven biogeochemical reactor in a temperate estuary"

_Biogeosciences, 2022_

## Author Comment (AC1)

**Response to Referee #1**

We thank the referee for the helpful and constructive comments on our manuscript. Below, we address how the individual issues that were raised will be addressed in the revised manuscript version.

*Line: 62*

*The significance of evaluation of nitrogen transformation along Elbe estuary under "intense summer oxygen depletion" is a little vague. Why do sample the water column under unusual condition? Please revise them for clearer description.*

We were indeed somewhat carried away describing the situation as so unusual and exceptional. It is indeed so that low oxygen conditions occur in summer in the Elbe /Port region, and that they can lead to fish kills and overall a bad ecological status for the river/ estuary. Our intention was two-fold: (1), to use this disruption of nitrogen cycling (as visible in nitrite accumulation) to assess isotope effects of nitrogen recycling processes in situ, and (2), to determine driving factors for biological turnover – which will hold regardless of low oxygen concentrations.

We will rephrase the intention of the study in the introduction to make this clearer.

*Line: 70*

*In the explanation of study site, there is a lack of information where an agricultural catchment area and areas of nutrient discharge (Line 71-73) exist. Where is the input of N ? Those information helps us to understand the interpretation of geochemical data along Elbe Estuary.*

This is true. The catchment is largely agricultural, especially in the Czech republic and the middle reaches of the River in Germany, which creates a high N input to the estuary. Diffuse input of N into the stream occurs mainly upstream of our sampling stretch via input from upstream tributaries. These receive N from agricultural (diffuse) sources. We will explain this in more detail in the revised version of the manuscript. As for the area of N-discharge, we were here referring to the mouth of the Estuary. We will clarify this in the revision.

*Line: 128*

*As for nitrate isotope analysis, how was nitrite removed from the nitrate samples? Some samples contained enough nitrite for isotope measurements. Thus, the presence of nitrite interfered with nitrate isotope measurements.*

Nitrite was measured separately using Stenotrophomonas nitritireducens, as outlined in the methods section. In these samples, the nitrate (or rather, nitrite + nitrate) isotope composition, as determined with Ps. aureofaciens, was corrected for the measured nitrite isotope value. We will add an explanation to the method section.

*Line: 150*

*It seems that both of ammonium oxidation and nitrite oxidation occur between stream km 641 and 656. Isotope compositions of nitrite could be affected by both of ammonium oxidation and nitrite oxidation. Do you consider the influence of ammonium oxidation on nitrite isotopes?*

It is true that both processes may occur in parallel between stream km 641 and 656. However, in this section, ammonium concentration is already low (3.5 µM at stream km 641, then 1.5 µm or less). Hence, we did not consider the isotope effect that ammonium

oxidation may have on nitrite isotopes. We actually assume that in this river section, ammonium assimilation gains in importance, because chlorophyll fluorescence increases. In this case, little ammonium would be oxidized to nitrite. We will insert a short explanation regarding low ammonium concentration, and will briefly assess this matter when discussing alternative ammonium sinks.

*Line:261*

*How did you calculate and conclude that the drop of oxygen isotope values in the harbor region was due to nitrification? What is the value of oxygen composition of river water ( $\delta_{18}O_{H2O}$) ?*

In this sampling, we did not measure the isotope value of Elbe water. However, as a large river, the Elbe has relatively little seasonal fluctuations in d18O_H2O, and outside the salinity gradient, the isotope value should be ~ -8‰ (Johannsen et al., 2008). We intentionally kept this statement qualitative, because the effects of nitrification on oxygen isotopes in nitrate are addressed elsewhere (Sanders et al., 2018). There, the authors find that the newly added nitrate has isotope values that are between -2.8 and -9.5‰, which is in the range of ambient water in the Elbe, and thus supports the relevance of nitrification. We will more clearly refer to this study in the revised version.

*Line:320*

*The authors constructed a total isotope mass balance by modeling. I understood the assumption of the model. However, the equation, parameters and calculation method were not presented in this text. Therefore, it is difficult to understand the conclusion that Total N fluctuations are driven by PN fluctuations. I'm not so familiar with this box model, but it seems better that the authors briefly explain equations and parameters in the box model and a modeling software in the method section or supplemental information.*

Going through the text again, we agree that the description of the calculation may appear a bit too brief. However, the initial description of a isotope box model probably was misleading. The calculations are actually rather based on a mass balance approach: Each point of measurements along the river is conceptualized as representing the homogenous composition of a parcel of water, with the sampling location as the center of the parcel. We consider four pools of N (PON, NO3-, NO2-, NH4+), and calculate the total N (TN) pool as:

[TN] = [PON] + [NO3-] + [NO2-] + [NH4+] $\qquad\qquad$ Eq S1

The mass weighted isotopic composition of total nitrogen (TN) is computed as:

$\delta^{15}N_{TN}$ = [$\delta^{15}N_{NO3}$ * [NO$_3^-$] + $\delta^{15}N_{NH4}$ * [NH$_4^+$] + $\delta^{15}N_{NO2}$ * [NO$_2^-$] + $\delta^{15}N_{PN}$ * [PN]]/

$\qquad$ [[NO$_3^-$]+ [NH$_4^+$] + [NO$_2^-$] + [PN]] $\qquad\qquad$ Eq. S2

We use this isotope mass balance model to examine downstream changes to total N and explore biogeochemical explanations for regions where observations are in violation of isotope mass balance. The description of the mass balance as such is already incorporated in the text in section 4.3.

Beyond this mass balance, and isotope mass balance, approach, we did not construct a box model. So, to avoid confusion, we will clearly refer to a mass balance approach (instead of referring to a box model) in the revised version. We will also add a supplement in which the above equations and assumptions are spelled out.

References

Johannsen, A., Dähnke, K., and Emeis, K.: Isotopic composition of nitrate in five German rivers discharging into the North Sea, Organic Geochemistry, 39, 1678-1689, 2008.

Sanders, T., Schöl, A., and Dähnke, K.: Hot Spots of Nitrification in the Elbe Estuary and Their Impact on Nitrate Regeneration, Estuaries and Coasts, 41, 128-138, 10.1007/s12237-017-0264-8, 2018.

---

## Author Comment (AC2)

Response to Referee #2

We thank the referee for the constructive and elaborate comments on the manuscript. Below, we address each issue individually, and explain the changes we will make to the revised manuscript to meet the reviewers criticism.

*Throughout, language such as unusual, exceptional, and unique are used to describe the conditions encountered, so how applicable are your results, just to these conditions? How often do these conditions occur? Are your findings applicable to the conditions seen in Sanders et al, 2018? It would be nice to see more comparisons drawn to this dataset.*

The most important difference between the cruises that are regarded in Sanders et al (2018) is that oxygen saturation is even lower (32% saturation in comparison to ~40% in the lowest depletion situation in Sanders et al. 2018). This leads to further elevated nitrite and ammonium concentration. However, we admit that the exceptionality of biogeochemical conditions during our cruise may be a bit overemphasized. Accumulation of ammonium and nitrite, and concurrent oxygen depletion, occurs regularly in the Elbe – only rarely so that the number of sampling points within the regions of ammonium and nitrite accumulation allows for computation of isotope effects, which probably tempted us to refer to the situation as a bit too exceptional. We will moderate the language use in the manuscript and insert a short paragraph in section 4.1 where we compare the data to Sander et al. (2018). This may also alleviate some issues regarding the nitrification rate assessment, as discussed below.

*Nitrification rates: more details are needed both in the methods and data interpretation.*

Nitrification rate measurements (and the intentional use of the long incubations, including potential caveats) are discussed in more detail in Sanders et al., 2018. Below, we will address the individual issues that were raised in more detail.

- *Nitrite and nitrate concentrations were stable, meaning all ammonium had been utilized?*

  Yes, that is the underlying assumption. Ammonium is utilized, and remineralization has apparently ended. Based on this approach, we aim to evaluate the nitrification potential of a given water sample, without addition of any extra substrate. We will refer to this briefly in the method description.

- *room temperature, was this close to insitu conditions?*

  Yes, under the given conditions in the Elbe, room temperature was indeed comparable to the water temperature in situ. The average water temperature in the sampling stretch was close to 24°C, and temperature in the labs is held at 22 +/- 2 degrees. We will add this to the method section.

- *14 days seems a long time and that bottle effects would be likely, was there any sign of this in the data? Exponential behavior for example? It would be beneficial to show some of this data, maybe in a supplement. How representative are these rates of insitu, as there seem to be a number of caveats, none of which are mentioned and there is also no comparison of the rates determined to those in the literature to put them in context, it is only mentioned that they are high.*

Indeed, nitrification rates are not addressed in detail in our study, as this was not the most important focus area. The incubation method is based on DINXXX, and is described in Sanders et al., 2018. We reckon that 14 days appear long, especially in comparison to frequently used techniques that are based on addition of labeled ammonium. However, the caveat of these methods is that ammonium is usually rapidly converted (or taken up) by any kind of micro-organisms, and our approach was precisely to address nitrification based on the material that was present in the water at the time of sampling.

Any rate determination in the lab will have its advantages and disadvantages, and a long-term incubation may overestimate remineralization and thus in-situ rates (see (Sanders and Laanbroek, 2018). We will, as mentioned above, compare the situation during our cruise to the data from Sanders et al. 2018, and will use this opportunity to briefly compare the nitrification rates. The measured rates are comparable, but slightly higher than average in Sanders et al., hence referred to later as "high". We will address this, but will refrain from a more detailed comparison to nitrification rates measured with other methods, because this is not the focus of our study and has been done previously by Sanders et al. (2018)).

- *you mention in the methods that ammonia and nitrite oxidation rates were determined but this is not mentioned in the results/discussion.*

In the manuscript, we refer to incubation rates as the average of all 4 incubations. Nitrite and ammonium oxidation rates were very similar, which is why we decided to present the overall nitrification rate as an average of all assessments. We will more clearly write this in the revised version of the manuscript to avoid confusion.

*Isotope mass balance box model: it is difficult to assess the outcomes of the model as no details are provided, equations, parameters etc, please provide this in the methods or supplement.*

We agree. The way it is currently phrased out in the manuscript, the calculations are not as clear as they could be. We address this topic as well in the response to reviewer #1. We actually used an isotope mass balance approach rather that an actual box model, so there are no model parameters we could spell out.

In the mass balance approach, we considered the four pools of N (PON, NO3-, NO2-, NH4+), and  calculate the total N (TN) pool as:

$$[TN] = [PON] + [NO3\text{-}] + [NO2\text{-}] + [NH4\text{+}]  \hspace{3cm} Eq\ S1$$

The mass weighted isotopic composition of total nitrogen (TN) is computed as:

$$\delta^{15}N_{TN} = [\delta^{15}N_{NO3} * [NO_3^-] + \delta^{15}N_{NH4} * [NH_4^+] + \delta^{15}N_{NO2} * [NO_2^-] + \delta^{15}N_{PN} * [PN]]/$$

$$[[NO_3^-\text{-}]+ [NH_4^+] + [NO_2^-] + [PN]]  \hspace{3cm} Eq.\ S2$$

We use this isotope mass balance model to examine downstream changes to total N and explore biogeochemical explanations for regions where observations are in violation of isotope mass balance. To make this point clear, we will add a short supplement with some details on the mass balance and with the above equation.

*Specific comments*

*Line 34 to 36: it is not clear how the second half of the sentence links to the first*

True. We will revise this. The line of thought was that oxygen conditions have improved, but this has in turn fueled nitrification, so that the nitrate load today remains high (despite reduction measures) and is at times even doubled in the estuary. We will make this clear in a revised version.

*Line 106: How was chlorophyll analyzed*

Chorophyll is measured by fluorescence with an on-line sensor that was part of the Ferrybox system (SCUFA Fluorometer, Turner Designs, San Jose, CA, USA). This is mentioned in line 82/83. We will insert a short reference to the sensor data later in the text.

*Line 126 / Nitrate Isotopes: There is no mention of a nitrite removal step, so are these actually N+N and not nitrate only? Please note the implications of this.*

The nitrate isotope values were measured with Pseudomonas aureofaciens, but they were corrected for the contribution of nitrite isotopes, which were measured independently with Stenotrophomonas nitritireducens. We will clarify this in a revision.

*Line 137: You note here that high concentrations were needed for isotope analysis of nitrite and ammonium, please include what concentrations needed to be greater than for isotopic analysis*

We usually measured samples that contained 2 μmol L-1 nitrite or ammonium, or more. We will add this to the text in the revised version.

*Line 225: For ammonium you use $\varepsilon_{amm}$ to represent the isotope effect for ammonium removal and then go on to discuss uptake and oxidation, which is great, but why not the same for nitrite? Here you assume it is just nitrite oxidation ($\varepsilon_{nitox}$), but highlight later in the manuscript a potential role for denitrification in this system (e.g. Line 372), which would also consume nitrite, what would be the implications of this for your calculated isotope effect?*

This is a good point. The main motivation to evaluate alternative ammonium sinks was the mismatch between $\Delta\delta$ ($\delta15N\_NH4 - \delta15N\_NO2$) and $\varepsilon15_{ammox}$ (line 274 - 289). We do not see indications for an unusual nitrite oxidation isotope effect in our study.

Additionally, throughout most of the sampling stretch, denitrification will most likely occur in sediments, where it will not affect d15N_NO2. The section of possible water column denitrification is stream km 618 – 635, where d15N-NO2 is relatively stable. The isotope effect of nitrite was calculated at decreasing nitrite concentration, from stream km 641 to 656. We have no indication for an additional sink process for nitrite in this stretch (in contrast to ammonium), so we did not evaluate the role of water column denitrification in this case. For clarity, we will insert a brief reference to this near the discussion of ammonium isotope effects.

*Figure 4 and associated text: it would be nice to see some errors on the calculated isotope effects.*

We will add measures of uncertainty of the slope in the figures and text in the revised version of the manuscript.

*Line 294 to 298: Across these lines, you discuss how nitrification scales / correlates with N content (%) and indicators of OM quality, where do I see this, you refer to Figure 4, but this is your isotope effects figure. These relationships need to clearly evident to support your conclusions.*

We thank the reviewer for notifying us on this mismatch. This was probably a remnant of a more lengthy discussion in a previous draft; we apologize for the mistake. Nitrification rates and N%(SPM) are correlated, see figure below. However, this is indeed not shown in Figure 4. We will correct this in a revision and refer to the actual correlation.

[Figure]

**Figure 1: Correlation of net nitrification rates and N% in suspended particulate matter.**

*Line 305 to 307 (and throughout this section): more explanation is needed for SPM reactivity, use the literature, for example, why does low C/N suggest its fresh and labile, references and details are needed for the reader to keep up with your line of thinking and confirm your conclusions.*

We will revise this paragraph and back it up with references to better guide the reader and support our conclusions. For fresh organic matter, we refer to the Redfield ratio, for which a C : N ratio of 6 -7 has been calculated (e.g., 6.6 in (Martiny et al., 2014)). With increasing remineralization, easily accessible N is used, and the C/N ratio increases (e.g., (Islam et al., 2019). We will add a short paragraph on OM reactivity in this section.

References:

Islam, M. J., Jang, C., Eum, J., Jung, S.-m., Shin, M.-S., Lee, Y., Choi, Y., and Kim, B.: C:N:P stoichiometry of particulate and dissolved organic matter in river waters and changes during decomposition, Journal of Ecology and Environment, 43, 4, 10.1186/s41610-018-0101-4, 2019.

Martiny, A. C., Vrugt, J. A., and Lomas, M. W.: Concentrations and ratios of particulate organic carbon, nitrogen, and phosphorus in the global ocean, Scientific data, 1, 140048, 10.1038/sdata.2014.48, 2014.

Sanders, T., and Laanbroek, H. J.: The distribution of sediment and water column nitrification potential in the hyper-turbid Ems estuary, Aquatic Sciences, 80, 10.1007/s00027-018-0584-1, 2018.

Sanders, T., Schöl, A., and Dähnke, K.: Hot Spots of Nitrification in the Elbe Estuary and Their Impact on Nitrate Regeneration, Estuaries and Coasts, 41, 128-138, 10.1007/s12237-017-0264-8, 2018.

---

## Author Response (AR1)

**Reply to the anonymous comments by reviewer #1 and #2 and the editor's comments**

We would like to thank both reviewers and editor for the constructive feedback on the manuscript. We now modified the manuscript according to the suggestions and hope that all issues that were raised could be solved.

A major concern of both reviewers was that the isotope mass balance model mentioned in the previous version of the manuscript was not well defined. As we outlined in the original response letter, referring to the calculations as a model was misleading. We actually performed mass balance calculations for the N-bearing substances, including their isotope composition, to track areas where the mass balance was violated. We now clearly refer to the calculations as a mass balance approach (e.g. lines 387, 414), and additionally added a supplement in which the underlying calculations are spelled out.

Both reviewers also noted that we refer to the hydrological situation in the Elbe during our study as so unusual, which indeed would raise the question of the study motivation. We modified the introduction to point out that oxygen minimum conditions, similar to the one we captured, occur regularly in summer, with the resulting ecological problems. However, these events are rarely captured in detail, and the spatial extent of nitrite and ammonium peaks in the harbor region gave us the opportunity to assess summer biogeochemical processes based on isotope effects, and with an isotope mass balance.

To make these points clear, we moderated the language use in the manuscript, and refer clearly to other studies that address oxygen conditions in the estuary (lines 35 – 38, 43 – 46, 70 – 74).

Following the suggestion of especially Reviewer #2 and the editor, we also revised the method section. We added methodological details throughout the section, and amended and largely simplified the description of rate calculations (lines 135 – 142). In detail, we added some text regarding the constraints of the method, referred more clearly to previous used of the method, and deleted the misleading section dealing with the separation of nitrite and ammonia oxidation.

Additionally, we made minor editorial edits to the discussion and conclusion section, and included a table in the supplementary material to show the changes in SPM content and quality. In the following, we now describe in detail which changes and modifications were made in response to the individual comments. Comments are in italics; our reply is in plain font.

**Reviewer 1**

*Line: 62 The significance of evaluation of nitrogen transformation along Elbe estuary under "intense summer oxygen depletion" is a little vague. Why do sample the water column under unusual condition? Please revise them for clearer description.*

→As outlined above (and in the original response letter), we now clearly contextualize the oxygen conditions in the Elbe, and rephrased the intention of our study to avoid confusion and to make clear that we could use this situation to unravel nitrogen cycling under summer conditions in the estuary (lines 35 – 38, 43 – 46, 70 – 74).

*Line: 70 - In the explanation of study site, there is a lack of information where an agricultural catchment area and areas of nutrient discharge (Line 71-73) exist. Where is the input of N ? Those information helps us to understand the interpretation of geochemical data along Elbe Estuary.*

→We added some information regarding diffuse and point sources of nitrogen (lines 78 – 81).

*Line: 128 - As for nitrate isotope analysis, how was nitrite removed from the nitrate samples? Some samples contained enough nitrite for isotope measurements. Thus, the presence of nitrite interfered with nitrate isotope measurements.*

→Nitrite was measured separately using Stenotrophomonas nitritireducens, as outlined in the methods section. In samples containing nitrite and nitrate, the nitrate isotope composition was determined by difference. We now mention this in the method section (lines 154 / 155)

*Line: 150 - It seems that both of ammonium oxidation and nitrite oxidation occur between stream km 641 and 656. Isotope compositions of nitrite could be affected by both of ammonium oxidation and nitrite oxidation. Do you consider the influence of ammonium oxidation on nitrite isotopes?*

→As we explain in the original response letter, we do not consider the effect of ammonium oxidation, because the ammonium concentration is already low in the section where both processes prevail, and should thus have little effect on the nitrite isotope composition. We now explain this briefly in the manuscript (lines 183 – 185).

*Line:261 - How did you calculate and conclude that the drop of oxygen isotope values in the harbor region was due to nitrification? What is the value of oxygen composition of river water (δ18OH2O) ?*

→The oxygen isotopes are not discussed in detail in our study, we refer here to previous study by Sanders et al (2018). We modified this section to point this out (lines 288 – 293). We did not include water isotope values, because we do not evaluate isotope changes of oxygen in detail in this present manuscript.

*Line:320 - The authors constructed a total isotope mass balance by modeling. I understood the assumption of the model. However, the equation, parameters and calculation method were not presented in this text. Therefore, it is difficult to understand the conclusion that Total N fluctuations are driven by PN fluctuations. I'm not so familiar with this box model, but it seems better that the authors briefly explain equations and parameters in the box model and a modeling software in the method section or supplemental information.*

→As outlined above, we now refer to our calculations as a mass balance approach rather than a model in section 4.3. We also added a supplement for further reference.

**Reviewer 2**

*Throughout, language such as unusual, exceptional, and unique are used to describe the conditions encountered, so how applicable are your results, just to these conditions? How often do these conditions occur? Are your findings applicable to the conditions seen in Sanders et al, 2018? It would be nice to see more comparisons drawn to this dataset.*

→As outlined above, we now present some more background information regarding the oxygen conditions, and moderated the language use. We also rephrased the study intention for clarity (lines 35 – 38, 43 – 46, 70 – 74). Later in the manuscript, we now compare our findings in some more detail to those by Sanders et al (2018), lines 288-291; 299 – 303.

*Nitrification rates: more details are needed both in the methods and data interpretation.*

→We revised this section in accordance with the reviewer's comments, including all bullet points that are mentioned separately. We specify implications of the used method now, clearly refer the reader to (Sanders and Laanbroek, 2018) for a method description, and included more details regarding the incubation. We also shortened and simplified the description of the rate calculation for clarity (lines 130-133; 135 – 143).

*Isotope mass balance box model: it is difficult to assess the outcomes of the model as no details are provided, equations, parameters etc, please provide this in the methods or supplement.*

→We agree. As outlined above, we now clarify that we use an isotope mass balance calculation rather than a model, and also included a supplement with mass balance calculations that the reader can refer to.

*Line 34 to 36: it is not clear how the second half of the sentence links to the first*

→We revised this section so that the linkage of oxygen and nitrification now is clear (lines 36 – 39).

*Line 106: How was chlorophyll analyzed*

→We now specify that chlorophyll fluorescence data are used as a quality indicator (117/118).

*Line 126 / Nitrate Isotopes: There is no mention of a nitrite removal step, so are these actually N+N and not nitrate only? Please note the implications of this.*

→We now mention that nitrate isotopes are determined by difference in cases where nitrite is present (lines 154/155)

*Line 137: You note here that high concentrations were needed for isotope analysis of nitrite and ammonium, please include what concentrations needed to be greater than for isotopic analysis*

→Done (lines 158; 168).

*Line 225: For ammonium you use $\varepsilon_{amm}$ to represent the isotope effect for ammonium removal and then go on to discuss uptake and oxidation, which is great, but why not the same for nitrite? Here you assume it is just nitrite oxidation ($\varepsilon_{nitox}$), but highlight later in the manuscript a potential role for denitrification in this system (e.g. Line 372), which would also consume nitrite, what would be the implications of this for your calculated isotope effect?*

→In the initial response letter, we explained that the main motivation to evaluate alternative ammonium sinks was the mismatch between $\Delta\delta(\delta15N\_NH4 - \delta15N\_NO2)$ and $\delta15_{ammox}$ (line 274 - 289). We now explain that the isotope effect for nitrite removal meets our expectations, making alternative sinks unlikely (lines 308 - 310).

*Figure 4 and associated text: it would be nice to see some errors on the calculated isotope effects.*

→We included the slope uncertainty in figure 4 and in the text

*Line 294 to 298: Across these lines, you discuss how nitrification scales / correlates with N content (%) and indicators of OM quality, where do I see this, you refer to Figure 4, but this is your isotope effects figure. These relationships need to clearly evident to support your conclusions.*

→We corrected this now and revised this paragraph (lines 333-339). We now explain that SPM quality is linked to nutrient turnover, and refer the reader to Fig. 2. To allow a more detailed evaluation of SPM quality, we also included a Table S1 in the supplementary material that shows the quality parameters we refer to.

*Line 305 to 307 (and throughout this section): more explanation is needed for SPM reactivity, use the literature, for example, why does low C/N suggest its fresh and labile, references and details are needed for the reader to keep up with your line of thinking and confirm your conclusions.*

→We inserted a paragraph that addresses OM reactivity and C/N ratios to better guide the reader, including the corresponding references (lines 345 – 350).

**Comments from the editor:**

*N sources: In your response to the first reviewer's comment (line 70), you focused on upstream "diffuse sources" (nonpoint sources). I wondered if you could also add some quantitative information about point sources such as wastewater effluents from big cities near and within the estuary.*

→We modified this paragraph and now mention diffuse as well as point sources (lines 79 – 81). Point sources usually do not play a significant role. An exception may be extreme rain events, in which a local input from the waste water treatment plant in the port of Hamburg may occur. However, this was not the case during our study.

*O2 level: In terms of, again, providing quantitative information, I thought you could better respond to the first comment of the second review on the extent of O2 depletion by providing the overall range of DO in the Elbe River (better citing papers that addressed the issue of O2 depletion in the same river system), or temporal variations of DO in the estuary if data are available.*

→We now provide more detail regarding the oxygen availability in the estuary to better introduce the study site and our study motivation (lines 36-38 and 43 – 46

*Lines 8-9: Do you mean "biogeochemical reactors that act to modify the loads and composition of nutrients transported to the coastal zone."?*

→Indeed. We changed this sentence (line 8/9).

*Lines 14-18: Please remove the very general background information (e.g., "estuarine biogeochemistry is governed by settling, resuspension, and remineralization of particulate matter"); instead provide "actual findings" that you want to highlight in the abstract. It should be noted that the abstract requires a substantial revision to provide the key findings and their implications.*

→We revised this abstract to make clear that these general pieces of information are actually true and relevant specifically for the Elbe estuary. Moreover, we modified the abstract in accordance with our findings (lines 15 – 18).

*Sampling and sample analysis: Please pay more attention to details during the revision. For example, "surface water samples" at what depth?; uniform descriptions of instruments (brand, company, country); QA/QC for water analyses (like blanks, replicates, reference materials,,,)*

→More detail was added to the methods section (e.g. lines 90; 110/111; 114; 120-127).

*Discussion also requires a substantial revision, because many general (background-like) descriptions are provided without clear linkages to the findings. Please consider a more focused discussion to highlight your points in a clearer way; for instance, when you begin a discussion section, you could articulate key findings in the context of your research questions.*

→We double-checked the discussion section. To streamline the discussion, we either removed some background statements (e.g. line 284, see also track change version of the manuscript), or, more frequently, linked them more clearly to our findings to highlight our points (e.g. lines 273; 276; 293; 304; 333-335). Additionally we added introductory sentences at the beginning of the discussion section to guide the reader (lines 268/269; 328/329; 408-410). . We paid attention to guide the reader to our findings throughout the discussion, and added more detailed background information (on SPM, oxygen, nitrification in the Elbe) whenever it was requested (see also response to the reviewers). If these revisions are not in accordance with the editor's expectations, we would be grateful for specific suggestions regarding changes. However, we do hope our changes are sufficient and in line with the editor's recommendations. We refrained from a more extensive restructuring because this would entirely change the manuscript and require resubmission.

*- Figs. 2-6: Please pay attention to details about axis titles and legends (e.g start the titles with upper-case letters; The initial "f" in Fig. 4 can be noted in the caption.*

→All axis titles in the respective figures were modified. We added a reference to "f" in the caption for Figure 4.

**References**

Sanders, T., and Laanbroek, H. J.: The distribution of sediment and water column nitrification potential in the hyper-turbid Ems estuary, Aquatic Sciences, 80, 10.1007/s00027-018-0584-1, 2018.

---

## Author Response (AR2)

**Reply to the comments by reviewer #1 and #2 and the editor's comments**

We thank the editor and the reviewers for the thorough review of the manuscript. We modified the manuscript, with particular focus on section 4.1 and 4.2, to clarify the connection of water column and sediment N processing to particulate matter concentration and quality. In line with this, we also replaced Figure 6 by a new version that hopefully makes it easier to follow the discussion on N sources and sinks derived from the budget calculation. In the following, we reply in detail to each individual comment by the editor / the reviewers.

- The main part of the abstract: Thanks for revising it in response to my suggestion, but I still find it difficult to figure out how you have derived your interpretations from the "actual findings". For instance, what are the relevant results for "the estuarine nitrogen budget of the Elbe estuary is governed by settling, resuspension, and remineralization of particulate matter"? On which finding did you base your interpretation that "the reactivity and concentration of particulate matter in the estuary is the main control of nitrogen isotope dynamics and of remineralization"?

→ As suggested, we have now revised the argumentation in the discussion to more strongly highlight the role of suspended matter in supplying substrates for nitrogen cycling (nitrification and denitrification). The main point is that organic matter remineralization fuels remineralization, so nitrification as well as oxygen consumption, and that settling and resuspension of particulate matter seem to be directly linked to these processes. We largely re-ordered the discussion, especially sections 4.1 and 4.2, to make these findings clearer.

- Section 3.1 & Fig. 2: You repeatedly refer to the "harbor (port) region" and its entrance in Fig. 2. I wondered if you could indicate the area or a key location on the figure, like by using some shaded area or vertical line(s).

→ Done. Additionally, we now mention the harbor region in the Figure Caption of Figure 1, where it is also indicated in the figure itself.

- As I requested earlier, please avoid general (introductory) statements in Discussion and focus on key findings and their implications in line with your research questions. For instance, the new first sentence doesn't fit into the first paragraph. Rather you can start with the second paragraph, and put the first paragraph at the end of the second one to explain how your approach contrasts with previous studies. The third and fourth paragraphs, which repeat background information described in Introduction, can be removed or incorporated into the introductory sentences.

→ We re-evaluated the discussion section and re-ordered large parts of the sections 4.1 and 4.2. In the introduction, we now included information on denitrification and its isotope effects to unburden the discussion section (e.g., lines 59 – 65).

- Another example of vague discussion point: "Based on C / N ratios and chlorophyll, we find that SPM is not equally reactive throughout the estuary. O2 saturation, an indicator for heterotrophic metabolism, reaches its minimum in the harbor region, after chlorophyll fluorescence decreased, suggesting active decomposition of limnic algal material (Fig. 2a)". I guess you meant "depletion of labile algal material" in the low-oxygen harbor region by "active decomposition of limnic algal material". However, this and the following sentence are confusing in the sense of active decomposition zone.

→ We refer to limnic material here, as it stems from the freshwater upstream (i.e., limnic) part of the estuary, upstream of our sampling stretch. We rephrased the section for clarity (lines 330

– 336). Algal material is decomposed, I.e., remineralized, in this section, which consumes oxygen. We avoided the term "Depletion", because it can, in the context of this manuscript, easily be misinterpreted as "isotope depletion", which we think would confusing to the reader.

- As the first reviewer pointed out, I would also like to ask you to pay more attention to enhancing the cohesiveness of many short paragraphs (in contrast to the very long first paragraph of 4.4). And a thorough proofreading would reduce grammatical errors such as inconsistent use of tense (just to select one example: From 0.2 µmol L-1 at stream km 618, concentrations "decreased: to < 2 µmol L-1 (1.7 +/- 0.6) and "remains" relatively stable along the estuary.), many present tenses used in Results and Discussion, and some awkward sentences like that appearing in Conclusions ("Ammonium assimilation competes with nitrification and produces reactive organic matter is produced that can in turn increase reactivity in the estuary.")

➔ We have thoroughly proof-read the results and discussion section to spot remaining errors and corrected these.

**Reviewer 1**

Firstly, I would like to thank the authors for the time and consideration they put into the revision and reviewer response, which I believe has resulted in an improved manuscript. However, while the overall message is clearer, clarity is still needed in some places.

In addition to this I urge the authors to think about the readability of their manuscript, currently the majority of the manuscript is short paragraphs of 2 to 3 sentences, resulting in the text lacking flow and cohesiveness.

➔ We have now thoroughly revised the discussion section according to the reviewer's and the editor's suggestions. Additionally, while we believe that 1 paragraph should represent one thought, we sought to adjust paragraph length accordingly.

Line 45: some additional explanation / references are needed here, why does low oxygen result in the accumulation of ammonium and nitrite?

➔ Nitrifiers are sensitive to environmental changes including availability of oxygen, which interrupts the usually smooth coupling of ammonium and nitrite oxidation (Heiss and Fulweiler, 2016). A negative correlation of nitrite and ammonium concentration with oxygen saturation has been observed previously (e.g.,(Jacob et al., 2016)). We have now reordered the paragraphs in the introduction and generally aimed to focus and disentangle it.

Line 67: The opening sentence of this paragraph where you refer to a key finding of Sanders et al, 2018 is totally disconnected from the following sentences where you introduce your study, I suggest you delete it or reword to incorporate it into the rest of the section.

➔ We agree that a deletion would have been an easy fix, but the sentence is needed to explain our study motivation. We added a connecting sentence to make this connection clearer to the reader while restructuring the introduction (now lines 37-39).

Figure 1 (same is true for Figures 2 and 3): throughout you use the term harbour region and I suggest you use this language here in figure 1 instead of or in addition to Hamburg port. For clarity it would also be beneficial to mark the harbour region onto Figures 2 and 3 (and maybe also in Figure 3 the region you use to determine ammonium and nitrite isotope effects)

➔ Done. We have replaced "port" with "harbor" throughout and shaded the area of interest in Figures 2 and 3.

Line 142: the fact that you saw no significant differences between treatments, needs to be commented upon. Was sufficient inhibitor used? 0.5uM seems very low when looking at Belser and Mays, 1980.

➔ True. The µM escaped all proof-readings, and we're glad this was finally spotted. We changed the concentration to mM. We also clarified the explanation. Nitrite oxidation was indeed inhibited in the amended incubations, but due to the fact that the rates are governed by ammonium oxidation, the slopes for increasing nitrite concentration (addition of chlorate) and increasing nitrate concentration (unamended incubations) are statistically indistinguishable. We chose to mention both treatments, because we were under the assumption that the use of 4 incubations would decrease the uncertainty associated with the rate measurements. We hope that the extended explanation helps to solve this issue. Otherwise, we can easily remove the reference to amended incubations from the manuscript, the calculated rates will not significantly change, and all conclusions will remain unaffected.

Line 236: I find this text misleading and confusing regarding where you are defining the harbour region, as currently written here it sounds like the harbour region is from 618 to 656km, which I don't think is the case. Please clarify.

➔ We have now rephrased this section (lines 234 – 236), the harbor region is indeed defined as 615 – 635, following previous studies of this region (Amann et al., 2015;Brase et al., 2017;Sanders et al., 2018). Oxygen consumption starts in the harbor region, but continues further downstream.

Line 273 and 276: If you want to use the words rare and unusual as you do in this paragraph you need to provide some context and compare / contrast to the literature to demonstrate this is the case.

➔ It is true (as also noted by reviewer 2) that summer oxygen depletion is not so rare. However, the magnitude and spatial extent is unusual compared to previous measurements. As such, we were uniquely able to determine isotope compositions in a sufficient number of samples and thereby derive isotope effects. This is usually not the case. We have now rephrased this section to put this into context and refer to Sanders et al. 2018 for comparison.

Line 308/309: See comment below on section 4.4 as well, this only makes sense once I get to the end of the manuscript. Here you need to comment directly on a potential role for sedimentary and water column denitrification.

➔ We have now included a paragraph on denitrification in the introduction section. (lines 59 – 65).

Line 338: 'where organic matter is fresh' this statement comes too early as you don't provide evidence for this until later in this section where you introduce the C/N results.

➔ We have re-ordered the entire section 4.2 and in parts rephrased it for clarification.

Line 343: What is your evidence for the statement remineralization is prominent, im actually not even sure what that means – this section is about determining an isotope effect and comparing it to the literature.

➔ This section was rewritten, the isotope bit was moved to 4.1, and the role of remineralization is now hopefully addressed more clearly in the revised section 4.2

Line 359: potentially even denitrifying, where is the evidence for this prior to this statement? Or at this point of the manuscript are you relying on the literature, if so a reference is needed.

➔ We have rewritten this section, removing the statement. It originally related also to the reactivity of SM, which now is discussed later in the discussion (lines 339 – 343).

Section 4.4 : it is only here that things finally fall into place for the reader, you need to introduce the ideas of water column and sedimentary denitrification earlier and there impacts on N isotopes (maybe in the introduction along with where in the estuary you would expect to find each based on the literature?), as prior to this you just use the term denitrification. This is confusing for the reader as you conclude that your isotope effect is representative of only nitrite oxidation as it is unaffected by additional nitrite sinks or sources (line 308), but then mention a number of times in the text a role for denitrification in the Elbe and it is not 100% clear in all cases where in the estuary you are referring to (e.g. line 359). Would a cross plot of 18O and 15N of nitrate not be informative with respect to picking these processes apart – this seems typical for nitrification / denitrification in other studies so would be good to highlight why this method is not relevant here?

➔ We now mention denitrification and the respective isotope effects in the introduction section. We also re-ordered the section 4.1, in which isotope effects are discussed, and added reference to denitrification here.
The use of a $\delta^{18}O/\delta^{15}N$ cross plot is typically more informative under circumstances exhibiting net $NO_3^-$ loss, and specifically highlighting how singular loss mechanisms may be overprinted by simultaneously occurring production. In the case presented here, there is an accumulation of $NO_3^-$ over time (so net production), and the $\delta^{18}O/\delta^{15}N$ cross plot, although strongly correlated, instead reflects a strong pattern of mixing between upstream sources (high values) and in situ $NO_3^-$ production by nitrification (lower values).

Line 435: a reference is needed to demonstrate that denitrification inside particles is feasible

➔ With this statement, we postulate the hypothetical role of water column denitrification. References for the plausibility are given in the following lines. However, we do not have direct evidence for water column denitrification. We have rephrased this section for clarity (lines 420 – 427) and have included a reference for anoxic processing in flocculate material (Klawonn et al., 2015).

Line 445 to 448: more explanation is needed here for the non-expert, and references are also needed, for example it should be highlighted how these ideas do / don't differ from km 640 to 655 where you also postulate a role for sedimentary denitrification.

➔ The main difference with respect to sediment denitrification further upstream is that the isotope mass balance requirements are not met in this section. We changed Figure 6 to better illustrate our reasoning in this section and rewrote parts of this discussion section. Furthermore, we also added some more detail to our explanation to make our assumptions clearer non-isotope-specialists (434 – 441).

Line 457: language, produce used twice

➔ Changed.

Line 462: 'alter the balance of nitrification and denitrification' it is not clear to me where this was directly discussed in the manuscript with respect to OM.

➔ We have now reworded the conclusion section, to emphasize that organic matter reactivity (via remineralization) directly affects denitrification and nitrification, and thereby the relative activity of the two processes (lines 452 – 454).

Reviewer #2

We thank the reviewer for the positive evaluation of the manuscript. We addressed the specific comment regarding "rare events" by rephrasing the respective sentence in the course of the revision. (Formerly line 273).

**References**

Amann, T., Weiss, A., and Hartmann, J.: Inorganic Carbon Fluxes in the Inner Elbe Estuary, Germany, Estuaries and Coasts, 38, 192-210, 10.1007/s12237-014-9785-6, 2015.
Brase, L., Bange, H. W., Lendt, R., Sanders, T., and Dähnke, K.: High Resolution Measurements of Nitrous Oxide (N2O) in the Elbe Estuary, Frontiers in Marine Science, 4, 10.3389/fmars.2017.00162, 2017.
Heiss, E. M., and Fulweiler, R. W.: Coastal water column ammonium and nitrite oxidation are decoupled in summer, Estuarine, Coastal and Shelf Science, 178, 110-119, https://doi.org/10.1016/j.ecss.2016.06.002, 2016.
Jacob, J., Sanders, T., and Dähnke, K.: Nitrite consumption and associated isotope changes during a river flood event, Biogeosciences, 13, 5649–5659, 10.5194/bg-13-5649-2016, 2016.
Klawonn, I., Bonaglia, S., Brüchert, V., and Ploug, H.: Aerobic and anaerobic nitrogen transformation processes in N2-fixing cyanobacterial aggregates, The ISME Journal, 9, 1456-1466, 10.1038/ismej.2014.232, 2015.
Sanders, T., Schöl, A., and Dähnke, K.: Hot Spots of Nitrification in the Elbe Estuary and Their Impact on Nitrate Regeneration, Estuaries and Coasts, 41, 128-138, 10.1007/s12237-017-0264-8, 2018.